# P2Y_2_ Receptor Signaling in Health and Disease

**DOI:** 10.3390/ijms26199815

**Published:** 2025-10-09

**Authors:** Fatemeh Salarpour, Jean Sévigny

**Affiliations:** 1Département de Microbiologie-Infectiologie et D’immunologie, Centres PROTEO et ARThrite, Faculté de Médecine, Université Laval, Québec City, QC G1V 0A6, Canada; fatemeh.salarpour@crchudequebec.ulaval.ca; 2Axe Maladies Infectieuses et Immunitaires, Centre de Recherche du CHU de Québec Université Laval, Québec City, QC G1V 4G2, Canada

**Keywords:** P2Y_2_ receptor, nucleotide receptors, extracellular nucleotides, calcium signaling, inflammation, tissue repair, neurodegenerative diseases, dry eye disease, cystic fibrosis, breast cancer

## Abstract

P2Y_2_ receptors are a subclass of G protein-coupled receptors activated by the extracellular nucleotides ATP and UTP. These receptors are widely expressed in multiple tissues—including the brain, lungs, heart, and kidneys—and play pivotal roles in inflammation, wound healing, and cell migration. Through coupling with various G proteins, P2Y_2_ receptors initiate diverse intracellular signaling pathways that mediate calcium mobilization, cytokine release, and cytoskeletal reorganization. Recent studies highlight their dual roles in health and disease. In physiological contexts, P2Y_2_ receptors contribute to immune modulation and tissue repair. In pathological conditions, they are implicated in Alzheimer’s disease by promoting non-amyloidogenic processing of amyloid precursor protein and in dry eye disease by enhancing mucin secretion while modulating ocular inflammation. They also influence chloride secretion and mucosal hydration in cystic fibrosis and contribute to inflammatory regulation and epithelial repair in inflammatory bowel disease. Additionally, P2Y_2_ receptors modulate breast cancer progression by regulating cell adhesion, migration, and matrix remodeling. Their involvement in blood pressure regulation via epithelial sodium channel modulation and their facilitative role in HIV-1 entry further underscore their clinical significance. These multifaceted functions position P2Y_2_ receptors as promising therapeutic targets for diverse diseases, warranting further investigation for translational applications.

## 1. Introduction

### 1.1. Nucleotide Receptors (P2X Receptors and P2Y Receptors)

Nucleotide signaling is a complex biological process involving cellular responses initiated by the binding of extracellular nucleotides to P2 receptors on the cell membrane [1]. These receptors, categorized into P2Y and P2X receptors, are widely distributed in most tissues and cells, reflecting their essential role in maintaining cellular function and communication throughout different biological systems [2].

More specifically, P2X receptors are ligand-gated cationic channels encoded by seven distinct genes (P2X1–P2X7), each producing a unique subunit. These subunits can assemble into functional homo- or heterotrimeric channels, greatly expanding the diversity of P2X receptor functions and enabling distinct, rapid responses to ATP, such as calcium influx and depolarization of the cell membrane [3]. In contrast, P2Y receptors are G protein-coupled receptors (GPCRs) that consist of eight members: P2Y_1_, P2Y_2_, P2Y_4_, P2Y_6_, and P2Y_11–14_. These GPCRs are activated by a variety of extracellular nucleotides, such as ATP, UTP, ADP, and UDP [4]. They are instrumental in mediating various physiological and pathophysiological processes, including but not limited to endothelial-mediated vasodilation [5], inflammation [6], cell differentiation, migration, and proliferation [7].

P2Y receptors are found in widespread locations across different tissues and organs, highlighting their crucial role in modulating cell and organ function by transmitting extracellular signals into intracellular responses [8,9]. In addition, the importance of these extracellular signaling substances and their receptors in various physiological and pathological processes has been increasingly recognized in recent years, with a growing body of research dedicated to elucidating their roles. Due to their broad functionality, signaling mediated by these receptors plays a crucial role in cellular responses and intercellular communication that regulate various aspects of both normal physiology and disease progression [9,10,11,12]. Among these receptors, P2Y_2_ receptors are particularly noteworthy due to their distinguishing features, including their activation by both ATP and UTP [13], involvement in key physiological processes such as inflammation and wound healing [14], widespread tissue distribution, and their association with diverse signaling pathways [15]. These attributes make them unique and important targets for potential therapeutic interventions, especially in conditions where modulation of P2Y_2_ receptors’ activity could influence disease outcomes.

### 1.2. Structure of the P2Y_2_ Receptor

P2Y_2_ receptors are GPCRs for extracellular nucleotides, characterized by a structure consisting of seven transmembrane helices, connected by three short extracellular loops and three intracellular loops of variable length. Recent cryo-EM structures of P2Y_2_ reveal the receptor adopts an active conformation even in the absence of a ligand, suggesting a mechanism for its constitutive activity [16]. This receptor architecture enables a consensus RGD integrin-binding domain in its first extracellular loop to effectively interact and bind to aVb3/5 integrins (Figure 1, red region), which is crucial for receptor-mediated cellular adhesion and signaling processes. In addition, two glycosylation sites are present in the extracellular N-terminus, which are thought to be involved in protein folding, stability, and receptor-ligand interactions. In transmembrane regions, basic residues located in the upper portions of the sixth and seventh transmembrane α-helices interact with the negatively charged phosphate groups of the nucleotide ligands (Figure 1, orange region), stabilizing ligand binding and promoting receptor activation [17,18]. The recent structural data also identified a unique lipid-binding pocket adjacent to Transmembrane Helices 3 and 4 that may contribute to allosteric regulation and receptor stability [16].

The intracellular loops of the P2Y_2_ receptor, the FLNa interaction site, regulate Gq protein binding and its subsequent activation, which is essential for initiating downstream signaling cascades (Figure 1). Moreover, its intracellular C-terminal domain contains two adjacent consensus PXXP motifs, which serve as SH3 domain-binding sites known to mediate interactions with SH3 domain-containing proteins, potentially playing a role in various signaling pathways. Additionally, there are potential phosphorylation sites located in the third intracellular loop, which may take part in receptor desensitization and internalization (Figure 1, yellow region), processes that are critical for regulating the receptor’s sensitivity and responsiveness to repeated stimulation [17,18]. A structural comparison with other P2Y family members further highlights the distinct G-protein coupling interface of P2Y_2_R, which accommodates both Gq and Go family proteins [16].

The P2Y_2_ receptor shows strong evolutionary conservation among vertebrates, with the rat coding sequence displaying 94% identity to that of the mouse and 84% to human orthologs [19]. Both ATP and UTP activate murine and human P2Y_2_ receptors with comparable potencies, exhibiting EC_50_ values in the range of 1.5–5.8 μM. However, ATP was approximately 10-fold less potent than UTP in inducing homologous receptor desensitization, with reported IC_50_ values of 9.1–21.2 μM for ATP versus 0.7–2.9 μM for UTP in the examined cell systems. Detailed analyses of desensitization kinetics revealed that the murine P2Y_2_ receptor was slightly more resistant to agonist-induced desensitization compared to its human counterpart [20].

### 1.3. P2Y_2_ Receptor Signaling

These receptors are coupled to their immediate effectors, which are heterotrimeric G-proteins, and function as guanidine exchange factors. In the inactive state, heterotrimeric G-proteins are presented in the cell as αβγ trimers, where the three subunits are bound together in a stable complex. Upon activation, the Gα-subunit, which binds GDP in its inactive state, undergoes a conformational change and is dissociated from the Gβγ dimer upon GTP binding. As a result of this activation, two functionally active effectors, Gα and Gβγ, emerge. Upon ligand binding, P2Y_2_ receptors can activate different classes of G proteins, such as Gq/11, Go, and G12. The specific G-protein that is activated depends on the cellular context and the type of receptor-ligand interaction involved. Responses to GPCR activation vary across cell types, partially due to the type-specific expression of effector proteins and the complex crosstalk occurring between various signaling pathways, which allows for a wide range of cellular responses to be finely tuned in different physiological and pathological settings [16,17]. For instance, in some cells, activation of P2Y_2_ receptors by certain ligands can lead to proinflammatory responses, while in others, it might promote tissue repair.

Activation of the Gq protein-coupled P2Y_2_ receptors leads to intracellular calcium release. This coupling activates PLC, which hydrolyzes the membrane lipid phosphatidylinositol 4,5-bisphosphate (PIP2) into two secondary messengers: inositol trisphosphate (IP3) and diacylglycerol (DAG). IP3 then diffuses through the cytoplasm and binds to IP3 receptors on the endoplasmic reticulum, triggering a cascade of events that result in the release of calcium ions stored in the ER into the cytosol [21,22,23]. Concurrently, DAG remains in the plasma membrane and, together with the elevated intracellular calcium, activates protein kinase C (PKC). The activation of PKC and the increase in intracellular calcium orchestrate various downstream cellular responses, including changes in gene expression, cell proliferation, migration, and the release of proinflammatory cytokines [24]. These multifaceted signaling events are pivotal in mediating a variety of cellular processes, such as inflammation, immune response modulation, and tissue repair. Furthermore, P2Y_2_ receptor activation significantly influences the synthesis and/or release of various bioactive molecules, including arachidonic acid (AA), prostaglandins (PG), and nitric oxide (NO). These molecules play important roles in inflammatory responses by serving as mediators that amplify or modulate immune system activities and promote tissue repair processes. Additionally, the modulation of ion channel activity through P2Y_2_ receptor signaling pathways highlights the critical role of the P2Y_2_ receptors in these physiological processes, especially in contexts where fine-tuning of cellular responses is necessary to maintain homeostasis or respond to injury and infection [25].

Additionally, P2Y_2_ receptors can activate both Go- and Gα12-dependent pathways, which in turn activate Rac and RhoA, respectively. Rac and RhoA are small GTPases that play pivotal roles in cytoskeletal dynamics, cell polarity, and cell movement. This signaling requires an interaction between the P2Y_2_ receptors and αvβ3-integrin, a key integrin that facilitates cell adhesion and migration by binding to the extracellular matrix components. These pathways influence the reorganization of the actin cytoskeleton, promoting cellular processes such as migration and morphological changes, which are critical in wound healing and development. Such processes are essential for the body’s response to injury, where cell migration and morphological adaptations are needed to close wounds, rebuild tissue, and manage inflammation effectively.

P2Y_2_ receptor-mediated activation of the disintegrin and metalloproteinase domain-containing protein 10 (ADAM10) and ADAM17 contributes to nucleotide-induced shedding of membrane-bound growth factors and subsequent epidermal growth factor receptor signaling. ADAM10 and ADAM17 are involved in the shedding of membrane-bound precursors of various growth factors, cytokines, and other signaling molecules, thereby regulating key physiological processes such as cell growth, differentiation, and inflammatory responses. The P2Y_2_ receptors undergo agonist-induced desensitization in several cell types, a phenomenon where the receptor becomes less responsive to continuous or repeated stimulation by an agonist [26] (Figure 2). Agonist-dependent GPCR desensitization is generally mediated by a family of GPCR kinases (GRKs 1–7), which phosphorylate residues in intracellular domains of the receptor and promote the binding of β-arrestins. The β-arrestins not only block further G protein coupling but also assist in receptor internalization into clathrin-coated pits, where the receptors can be either recycled back to the plasma membrane or targeted for degradation. This process helps regulate receptor availability and sensitivity, maintaining cellular responsiveness to external signals [18,27,28].

### 1.4. Physiological Functions of the P2Y_2_ Receptor

P2Y_2_ receptors are extensively expressed across various organs and tissues, including the brain, lungs, heart, and kidneys. These receptors play a crucial role as sensors of their natural ligands, ATP and UTP, which can be released, for example, from apoptotic cells. In turn, P2Y_2_ receptors mediate the recruitment of immune cells to clear cellular debris, highlighting their key role in maintaining tissue homeostasis [29]. Recent studies have expanded our understanding of P2Y_2_ receptor function beyond merely debris clearance by phagocytes. It has become evident that P2Y_2_ receptor-mediated immune cell recruitment encompasses a broader range of activities. For instance, P2Y_2_ receptor expression has been identified in a variety of immune cell types, including dendritic cells, eosinophils, B and T lymphocytes, macrophages, mast cells, monocytes, neutrophils, and natural killer (NK) cells [1,30,31,32,33,34,35,36,37,38].

Moreover, P2Y_2_ receptors have been implicated in the transactivation of growth factor receptors, which enhances cell proliferation and upregulates immune cell adhesion molecules, such as VCAM-1 [24,33,34,39,40]. This indicates that P2Y_2_ receptors are not only involved in immune cell recruitment but also play a significant role in modulating immune responses and cellular interactions. Given the wide array of signaling pathways and biological processes controlled by P2Y_2_ receptors, these receptors are implicated not only in normal physiology but also in the pathogenesis of numerous diseases. The following section highlights their pathological roles in specific disorders.

## 2. Pathological Roles of the P2Y_2_ Receptor

### 2.1. Neurological Disorders: Alzheimer’s Disease

The etiology of Alzheimer’s disease remains poorly understood; however, it is characterized by the progressive degeneration of neurons and the loss of synaptic connections in the cerebral cortex and specific subcortical regions. This degeneration is linked to the accumulation of pathological protein deposits in the brain, specifically β-amyloid forming senile plaques and hyperphosphorylated tau protein forming neurofibrillary tangles [35,41,42].

Neurotoxic β-amyloid is produced through the activity of β-secretase and γ-secretase, which cleave amyloid precursor protein (APP). β-amyloid induces free-radical reactions and inflammation, ultimately leading to neuron death and the development of dementia. In contrast, the activity of α-secretase results in the creation of soluble secreted amyloid precursor protein-α (sAPP-α), which has neuroprotective and neurotrophic properties [36].

In 2009, the group of G.A. Weisman reported increased expression and activity of the P2Y_2_ receptor in rat cortical neurons in response to IL-1β. Under normal physiological conditions, the P2Y_2_ receptor is expressed at low levels and remains unresponsive to UTP. However, upon activation by UTP, these receptors stimulate the release of soluble amyloid precursor protein-α (sAPP-α) through a mechanism involving ADAM10/17 and phosphoinositide 3-kinase (PI3K). Additionally, signaling through extracellular signal-regulated kinase 1/2 (ERK1/2) and phosphoinositide 3-kinase (PI3K) may contribute indirectly to the regulation of APP processing [43].

Neuronal P2Y_2_ receptors also participate in regulating the phosphorylation of the cytoskeletal protein cofilin, which ultimately stabilizes the outgrowth of neurites and dendritic spines. This represents another neuroprotective mechanism involving this receptor in the context of Alzheimer’s disease [44]. P2Y_2_ receptors also mediate the release of arachidonic acid and activation of cyclooxygenase type 2 in astrocytes, which mediates inflammation in Alzheimer’s disease [45,46,47].

### 2.2. Ocular Disorders: Dry Eye Disease

Inflammation is a potential factor in ocular surface damage in dry eye disease (DED) [48]. P2 receptor subtypes are closely linked to the activation of inflammatory vesicle complexes and the subsequent release of proinflammatory cytokines, which contribute to the inflammatory response in the ocular surface. P2Y_2_ receptors are expressed in the corneal epithelium and endothelium, where they play a critical role in mediating calcium ion (Ca^2+^) mobilization upon activation by ATP or UTP. This Ca^2+^ mobilization activates downstream signaling pathways that promote the release of proinflammatory cytokines such as interleukin-8 (IL-8), contributing to ocular surface inflammation [49,50].

Despite this proinflammatory role, P2Y_2_ receptors also exhibit protective effects on the ocular surface. Their activation enhances mucin secretion from conjunctival goblet cells [51,52], thereby forming a lubricating mucus layer that is essential for ocular surface hydration and defense. Mucins help maintain epithelial barrier integrity, reduce pathogen adherence, and facilitate clearance of debris and microorganisms [53,54]. Additionally, P2Y_2_ receptor activation can inhibit apoptosis via ERK-p90RSK signaling and suppress the NF-κB pathway, which further helps protect the ocular surface from inflammatory damage [47].

This apparent contradiction, proinflammatory in some contexts, protective in others, has led to some confusion. However, clinically, P2Y_2_ receptor agonists such as diquafosol, a uridine nucleotide analog, are being used to treat dry eye disease [55,56], capitalizing on their ability to stimulate tear fluid and mucin secretion [57]. Diquafosol tetrasodium is approved in several countries for DED treatment, and studies have shown that it improves tear film stability and ocular surface health, particularly in cases with mucin deficiency [58].

In conclusion, although P2Y_2_ receptor activation can contribute to inflammation under certain conditions, its overall effect in the context of dry eye disease is beneficial. Agonists like diquafosol are currently used in therapy, making P2Y_2_ receptors a viable and promising target in the management of dry eye disease.

### 2.3. Respiratory Disorders: Cystic Fibrosis

P2Y_2_ receptors expressed on the alveolar epithelium play a crucial role in pulmonary homeostasis by regulating surfactant secretion. They promote the secretion of chloride (Cl^−^) and bicarbonate ions (HCO_3_^−^) while simultaneously inhibiting sodium (Na^+^) absorption via the epithelial sodium channel (ENaC) [59]. In addition to their epithelial functions, P2Y_2_ receptors are also expressed on immune cells, where they are essential for host defense. For instance, ATP released during cell injury acts as a “find-me” signal for monocytes, facilitating the clearance of apoptotic debris via P2Y_2_ receptor-mediated signaling [60]. These combined actions underscore the dual role of P2Y_2_ receptors in both preserving epithelial function and coordinating immune responses within the lung.

Cystic fibrosis is a genetic disease caused by mutations in the CFTR gene, leading to widespread defects in epithelial ion transport. These mutations result in the production of a dysfunctional CFTR channel, which severely impairs the regulation of Cl^−^ and sodium Na^+^ ions across epithelial cells. The most severe and life-threatening complications of cystic fibrosis arise from aberrant mucus production, which leads to obstruction and infection of the airway and intestinal lumens.

In the respiratory and digestive tracts, mucus is normally secreted to protect and lubricate the epithelial surfaces. However, in patients with cystic fibrosis, the defective CFTR protein leads to a failure in chloride ion transport. Mucin granules, which are key components of mucus, are released via Ca^2+^-dependent exocytosis following the interaction of agonists such as ATP and UTP with the P2Y_2_ receptors on goblet cells. These agonists activate the P2Y_2_ receptors, which are coupled to the Gαq/11 protein, leading to the promotion of Cl^−^ secretion from airway epithelium and mobilization of intracellular Ca^2+^ stores [61,62,63].

In the context of cystic fibrosis, defective Cl^−^ transport causes chloride ions to become trapped inside the cells. Without proper movement of Cl^−^ out of the cells, water cannot hydrate the cellular surface. This lack of hydration leads the mucus covering the cells to become thick and sticky, rather than thin and fluid. The thickened mucus causes many of the symptoms associated with cystic fibrosis, including chronic respiratory infections, persistent coughing, and digestive difficulties [64].

P2Y_2_ receptor agonists such as denufosol promote airway hydration by stimulating chloride and fluid secretion; however, denufosol failed to demonstrate clinical efficacy in cystic fibrosis patients in late-stage trials [57].

### 2.4. Inflammatory Disorders: Inflammatory Bowel Disease

Inflammatory bowel diseases (IBD), which include Crohn’s disease and ulcerative colitis, are chronic inflammatory conditions that lead to various complications. Ulcerative colitis is limited to the colon and is characterized by inflammation restricted to the mucosa, whereas Crohn’s disease can affect any part of the gastrointestinal tract and involves transmural inflammation [65]. The team of F.P. Gendron in Sherbrooke reported that P2Y_2_ receptor expression is increased in the colon of IBD patients and in the dextran sulfate sodium (DSS)-induced acute murine model of colitis [66,67].

Given their increased expression in both human IBD and experimental colitis, attention has turned to the functional role of P2Y_2_ receptors in intestinal inflammation. Emerging evidence indicates that, in addition to their contribution to inflammatory signaling, P2Y_2_ receptors may also participate in promoting mucosal repair and regeneration. In a 2013 study, Émilie Degagné and colleagues demonstrated that stimulation of P2Y_2_ receptors in intestinal epithelial cells triggers the release of prostaglandin E_2_ (PGE_2_). This response appears to serve a dual function. During the acute stage of inflammation, P2Y_2_ receptor-mediated activation of COX-2 and subsequent PGE_2_ production can amplify the inflammatory reaction in the intestinal mucosa—potentially contributing to tissue damage—while simultaneously restricting the translocation of foreign substances into the systemic circulation and initiating early tissue repair mechanisms. In the subsequent resolution phase, these mediators may promote regeneration by enhancing intestinal epithelial cell proliferation and strengthening barrier integrity [68].

Beyond PGE_2_-mediated effects, P2Y_2_ receptor activation engages additional signaling pathways that further boost mucosal healing. One such pathway involves the RGD motif in P2Y_2_ receptors, which binds integrins and promotes epithelial restitution through cytoskeletal reorganization [69].

### 2.5. Oncology: Breast Cancer

Activation of the P2Y_2_ receptors initiates a cascade of downstream signaling pathways that are heavily involved in various aspects of cancer biology, including tumorigenesis, metastasis, and the emergence of resistance to chemotherapeutic agents. Among these pathways are the activation of the MAPK signaling cascade, regulation of gene expression via the transcription factor AP1, transactivation of the epidermal growth factor receptor through the shedding of growth factors, and signaling through the Rac and Rho GTPase cascades. Each of these pathways plays a crucial role in the complex processes that underlie cancer progression and treatment resistance [28].

Breast cancer is the third most common cancer worldwide, after lung and colon cancer, and is the leading cause of cancer-related death in women [70]. Within the context of cancer, nucleotides are released not only by tumor cells themselves but also by host cells within the tumor microenvironment. The actions of these nucleotides significantly impact both the activation and migration of immune cells, as well as the growth and proliferation of tumor cells [71,72]. For example, cell adhesion molecules, such as intercellular adhesion molecule-1 (ICAM-1) and vascular cell adhesion molecule-1 (VCAM-1), play critical roles in cancer progression. ICAM-1 supports firm adhesion of circulating tumor cells to the vascular endothelium, facilitating intravasation and extravasation, whereas VCAM-1 promotes tumor cell survival, invasion, and colonization at distant sites through interactions with integrins on immune and stromal cells. Their upregulation is linked with enhanced metastatic potential and poor prognosis in breast cancer [73,74]. P2Y_2_ receptor activation in endothelial cells has been linked to the regulation of these adhesion molecules. Specifically, it increases the expression of ICAM-1 and VCAM-1 and phosphorylates vascular endothelial cadherin in breast adenocarcinoma [75].

In another study conducted in 2017, Zhang and colleagues found that P2Y_2_ receptor-targeting shRNAs attenuated extracellular ATP-induced migration and invasion of MDA-MB-231 human breast cancer cells, likely through disruption of P2Y_2_ receptor-mediated β-catenin signaling [76]. Furthermore, P2Y_2_ receptors may promote cancer cell metastasis by enhancing lysyl oxidase activity, which catalyzes the cross-linking of extracellular matrix proteins, such as collagen and elastin, during the formation of a premetastatic niche. Hypoxic conditions within the tumor microenvironment have been shown to promote ATP release from MDA-MB-231 cells, leading to subsequent P2Y_2_ receptor activation, which in turn induces lysyl oxidase secretion and collagen cross-linking [77].

### 2.6. Cardiovascular Disorders: Blood Pressure Regulation

Cardiovascular disease is the leading cause of death both in the United States and globally, accounting for about 20 million deaths annually. Hypertension is a significant risk factor for cardiovascular disease, with the risk escalating as blood pressure rises. In industrialized nations, close to 90% of individuals will develop hypertension during their lifetime [78,79,80].

Recent studies have shown that activation of P2Y_2_ receptors in the kidney is linked to increased sodium excretion and decreased blood pressure. Within the kidneys, epithelial sodium channel (ENaC) is expressed in principal cells, which are localized to the late distal convoluted tubule, the connecting tubule, and the collecting duct system [81]. ENaC activity in principal cells is regulated by peptide hormones (e.g., arginine vasopressin) [82], mineralocorticoid steroid hormones (e.g., aldosterone), and paracrine factors such as ATP. ATP decreases ENaC activity by stimulating luminal P2Y_2_ receptors in principal cells [83,84,85,86,87,88,89]. This activation of P2Y_2_ receptors stimulates PLC via Gq signaling, promoting the hydrolysis of membrane phosphatidylinositol PIP2. ENaC activity is dependent on membrane PIP2 levels such that decreases in membrane PIP2 levels decrease ENaC activity [89,90,91,92,93]. This suggests that P2Y_2_ receptors play a crucial role in regulating blood pressure through their effects on sodium handling in the kidneys. ENaC activity is elevated in *P2ry2*-knockout mice and closely resembles the increased blood pressure observed in humans with gain-of-function mutations in ENaC [83,84,94,95,96].

Recently, Soares AG reported on the necessity and sufficiency of P2Y_2_ receptor signaling in renal principal cells. This study, conducted in 2023, focused on the effects on ENaC activity, sodium excretion, and blood pressure. The research utilized principal cell-specific *P2ry2*-knockout mice and Gq-DREADD–knockin mice. They have demonstrated that inhibition of ENaC activity in response to P2Y_2_ receptor–mediated Gq signaling lowered blood pressure by increasing renal sodium excretion. Thus, P2Y_2_ is an emerging target capable of lowering elevated blood pressure, and its dysfunction may contribute to certain forms of hypertension [89].

In addition, an integrin-binding domain (Arg-Gly-Asp) in the first extracellular loop of P2Y_2_ receptors mediates its association with aVb3 integrins, enabling the coupling of P2Y_2_ receptors to G0 but not Gq proteins. This novel P2Y_2_ receptor/αVβ3 integrin interaction was found to regulate UTP-induced actin cytoskeletal rearrangements and promote cell migration. Since leukocyte infiltration and migration are key processes involved in atherosclerosis, these findings suggest that P2Y_2_ receptors may represent a novel target for reducing arterial inflammation associated with cardiovascular disease [45].

### 2.7. Infectious Diseases: HIV-1

P2Y_2_ receptors have been shown to play a significant role in mediating immune responses to pathogenic infections, serving as crucial components of the body’s defense mechanisms against a variety of pathogens [97]. In the context of viral infections, P2Y_2_ receptors are particularly important as they facilitate the entry of viruses into host cells, exemplified by their role in the infection process of human immunodeficiency virus (HIV). A study conducted by Séror in 2011 elucidated the complex mechanism through which HIV utilizes P2Y_2_ receptors for its entry and propagation within host cells [98]. The process begins with the binding of HIV envelope glycoproteins (Env) to CD4 and CXCR4 receptors on the surface of target cells. This interaction triggers the release of ATP through pannexin-1 hemichannels. The extracellular ATP then interacts with P2Y_2_ receptors on the host cell membrane, leading to the activation of proline-rich tyrosine kinase 2. This kinase activation is coupled with transient plasma membrane depolarization, a critical step that facilitates the fusion of viral membranes, which express Env, with the host cell membranes containing CD4 and the appropriate chemokine co-receptors [99].

The importance of P2Y_2_ receptors in this process is underscored by the observation that the loss of these receptors results in a significant reduction in Env-triggered syncytium formation, a hallmark of HIV infection, where multiple host cells fuse together. This syncytium formation is critical for viral replication and the spread of infection within the host. The reduction in syncytium formation due to the absence of P2Y_2_ receptors underscores their pivotal role in facilitating the fusion of infected cells, a mechanism that not only aids in the efficient replication of the virus but also facilitates evasion of the host’s immune response. By promoting the fusion of host cells, P2Y_2_ receptors contribute to the creation of multinucleated cells that can harbor large amounts of viral particles, thereby enhancing the virus’s ability to spread and persist within the host. The ability of P2Y_2_ receptors to influence such a critical aspect of HIV pathology suggests that they could be key targets for therapeutic interventions aimed at limiting the spread of the virus and potentially controlling the progression of the infection [91].

Furthermore, analyses of lymph node and frontal cortex biopsies from patients with HIV have revealed elevated levels of P2Y_2_ receptor expression, suggesting a potential upregulation of these receptors in response to HIV infection. This upregulation may reflect the virus’s exploitation of P2Y_2_ receptors to enhance its replication and persistence within the host. Additionally, overexpression of P2Y_2_ receptors in host cells significantly enhances the fusion efficiency of Env-expressing cells with CD4+ CXCR4+ T cells, further facilitating the viral life cycle. The increased expression of P2Y_2_ receptors not only boosts the fusion of viral and host cells but also potentially accelerates the spread of the virus to new cells, thereby exacerbating the progression of the infection. This evidence underscores the critical role of P2Y_2_ receptors in HIV pathogenesis, suggesting that these receptors could be pivotal targets for therapeutic strategies aimed at disrupting the viral life cycle and limiting the disease’s impact on the immune system. The correlation between P2Y_2_ receptor expression levels and the efficiency of viral fusion highlights the receptor’s potential as a biomarker for disease progression, as well as a possible target for interventions designed to mitigate the severity of HIV infection [98]. A summary of diseases and pathological conditions associated with P2Y_2_ receptor signaling is presented in Table 1.

## 3. Conclusions

In conclusion, P2Y_2_ receptors play important functions in both physiological and pathological processes, modulating inflammation, wound healing, and cell migration. They are implicated in a broad spectrum of diseases such as Alzheimer’s disease, dry eye disease, and cystic fibrosis. Their role in inflammatory diseases, particularly in IBD, is notable, as they contribute to intestinal epithelial repair and immune modulation. Additionally, P2Y_2_ receptors influence cancer progression, such as in breast cancer, and play a role in blood pressure regulation, and facilitate HIV infection. Taken together, these diverse functions underscore P2Y_2_ receptors as promising therapeutic targets across multiple disease areas.

## Figures and Tables

**Figure 1 ijms-26-09815-f001:**
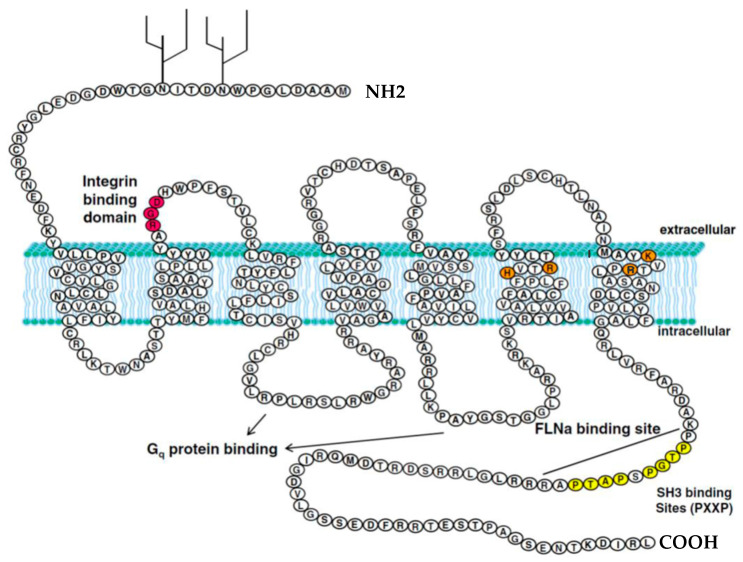
P2Y_2_ receptor structure and domains in humans. Inspired by Figure 1 in Peterson, Troy S et al., 2010. The P2Y_2_ receptor is a G protein-coupled receptor (GPCR) that spans the membrane seven times and specifically responds to extracellular nucleotides. It is equally activated by ATP and UTP, and its expression is often enhanced under conditions of cellular stress or tissue injury. Key structural features include a conserved RGD motif for integrin binding (red), positively charged residues that participate in ATP/UTP interaction (orange), two PXXP motifs that serve as SH3 domain-binding sites (yellow), the FLNa interaction site, intracellular loops involved in Gq protein coupling, and two glycosylation sites on the N-terminal region [18]. RGD motif: Arginine–Glycine–Aspartic acid motif, PXXP motifs: Proline–x–x–Proline motif, SH3: Src Homology 3, FLNa: Filamin A.

**Figure 2 ijms-26-09815-f002:**
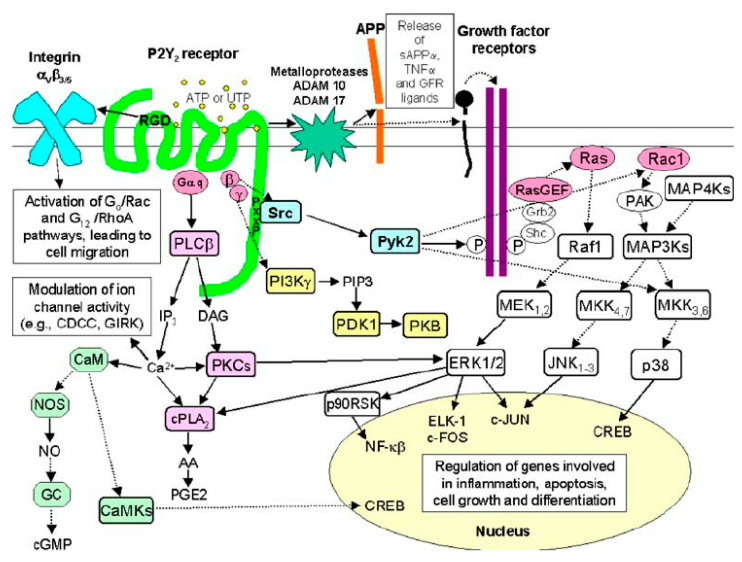
Intracellular signaling pathways are activated by the P2Y_2_ receptor. Upon ligand binding, P2Y_2_ activates various G-protein subtypes, including Gq/11, Go, and G12, depending on the cell type and context. Gq activation leads to stimulation of phospholipase C (PLC), which hydrolyzes PIP2 into IP3 and DAG. IP3 promotes calcium release from the endoplasmic reticulum, while DAG, in conjunction with elevated calcium, activates protein kinase C (PKC), initiating downstream responses like cytokine release and gene expression changes. Additionally, P2Y_2_ receptor signaling can trigger the release of inflammatory mediators such as arachidonic acid, prostaglandins, and nitric oxide. Activation of Go and G12 pathways results in the stimulation of small GTPases Rac and RhoA, which regulate cytoskeletal remodeling and cell migration via interactions with αvβ3 integrin. P2Y_2_ receptors also engage ADAM10 and ADAM17, facilitating the shedding of membrane-bound growth factor precursors and subsequent EGFR signaling. Receptor desensitization is mediated by GRKs and β-arrestins, which regulate receptor availability through internalization and recycling or degradation. Inspired by Figure 2 in reference Peterson, Troy S et al., 2010 [18]. PIP2: Phosphatidylinositol 4,5-bisphosphate, IP3: Inositol 1,4,5-trisphosphate, DAG: Diacylglycerol, ADAM: A Disintegrin and Metalloproteinase, EGFR: Epidermal Growth Factor Receptor, GRKs: G protein–coupled receptor kinases.

**Table 1 ijms-26-09815-t001:** Summary of diseases and pathological conditions associated with P2Y_2_ receptor signaling. The table highlights the main signaling pathways, molecular mechanisms, and physiological or pathological outcomes implicated in neurological, ocular, respiratory, inflammatory, oncological, cardiovascular, and infectious disorders.

Disease/Condition	Mechanisms Involving P2Y_2_	Physiological/Pathological Effects
Neurological: Alzheimer’s disease	↑ P2Y_2_ expression in neurons (IL-1β induced); ADAM10/17 activation → ↑ sAPP-α release; ERK1/2 and PI3K signaling; regulation of cofilin phosphorylation; arachidonic acid release in astrocytes	Neuroprotection via sAPP-α and neurite stability; inflammation via COX-2 and AA/PG release
Ocular: Dry Eye Disease	Ca^2+^ mobilization in corneal epithelium; cytokine release (IL-8); mucin secretion from goblet cells; ERK-p90RSK and NF-κB suppression	Dual role: proinflammatory vs. protective; clinical use of diquafosol to stimulate tears and mucins
Respiratory: Cystic Fibrosis	P2Y_2_ on epithelium → Cl−/HCO_3_− secretion and ENaC inhibition; ATP/UTP-triggered mucin release; immune cell signaling	Promotes airway hydration and mucus clearance; denufosol (agonist) failed in clinical trials
Inflammatory: Inflammatory Bowel Disease	↑ P2Y_2_ in colon of IBD patients and DSS-colitis mice; COX-2–PGE_2_ pathway; integrin binding (RGD motif) → cytoskeletal reorganization	Amplifies inflammation (acute); promotes epithelial repair and barrier regeneration (resolution phase)
Oncology: Breast Cancer	MAPK and AP-1 signaling; EGFR transactivation via ADAMs; Rac/Rho activation; regulation of ICAM-1 and VCAM-1; β-catenin signaling; lysyl oxidase induction under hypoxia	Promotes tumor cell migration, invasion, adhesion, metastasis, and chemoresistance
Cardiovascular: Hypertension & Atherosclerosis	Renal ENaC inhibition via PLC/PIP2 signaling → natriuresis & ↓ BP; αVβ3-integrin interaction → actin remodeling and leukocyte migration	Blood pressure regulation: P2Y_2_ dysfunction linked to hypertension; possible target for vascular inflammation
Infectious: HIV-1	ATP release via pannexin-1 → P2Y_2_ activation → proline-rich tyrosine kinase 2 activation → membrane depolarization → viral fusion (Env–CD4/CXCR4 mediated)	Facilitates viral entry, syncytium formation, replication, and spread; receptor upregulated in HIV patients

Symbols: → indicates causal or signaling direction; ↑ indicates increase or activation; ↓ indicates decrease or inhibition.

## Data Availability

No new data were created or analyzed in this study. Data sharing is not applicable to this article.

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
