# Peer review of "P2Y2 Receptor Signaling in Health and Disease"

_ijms, 2025, doi:10.3390/ijms26199815_

Round 1

Reviewer 1 Report

Comments and Suggestions for Authors

This is an interesting review focused in the relevance of P2Y2 receptor in some systems, the manuscript is of general interest, it is well presented and organized  and has updated information.

I have only some comments

  • According to IUPHAR, abbreviating receptors names is not recommended, I suggest changing the term P2Y2Rs to “P2Y2 receptors”. In analogy, this observation also applies to  “P2Rs” line 57, “P2XRs” line 57 and in general, to the entire manuscript.
  • Line 60 says: “P2XRs are ligand-gated cationic channels, comprising seven mammalian subtypes: P2X1–7Rs, each subtype triggering distinct yet rapid responses to ATP”. This is imprecise. There are seven genes encoding seven P2X subunits; however, a functional channel can be homo- or heterotrimeric, increasing the diversity of functional P2X receptor. Please revise this paragraph.
  • In section “Structure of the P2Y2 receptor”, what about differences or similarities between receptor from different species in terms of amino acid identity, pharmacology, etc?
  • The sentence “This receptor-mediated cytoskeletal reorganization is critical in mediating inflammation, immune responses, and tissue repair as well” , in lines 142-143, repeats the meaning of the previous sentence.
  • P2Y2 receptor phosphorylation and desensitization is mentioned in the review (lines 150-155), however the original paper demonstrating P2Y2 receptor phosphorylation is not cited (doi: 10.1007/s11010-005-8050-5), it should be cited.
  • On line 212, the original article must be cited instead of 39.
  • Lines 297 to 301 describe the role of P2Y2 receptor regulating ICAM-1 and VCAM-1, but the role of these adhesion molecules in growth and metastasis is not clear, please explore this topic in more detail.
  • A table or picture summarizing the information in the text can be a smart query tool.

Reviewer 2 Report

Comments and Suggestions for Authors

The review „ P2Y2 Receptor Signaling in Health and Disease” by Salarpour and Sevigny highlights the P2Y2 receptor and the role in very relevant disease conditions like AD, HIV infection, cancer and blood pressure regulation. The structure of the review is clear and it is very well written. I have only minor points.

Line 60-64 please add references

Line 80 please include recent details about structural data e.g. doi.org/10.1038/s41421-025-00797-x

Fig1. please highlight N- and C-terminus and add the species

Fig2. Can you improve quality by increasing resolution

Line  319 Arginine is not a peptide. What reference is supporting arginines role in ENaC modulation?
